# Text3DSAM: Text-Guided 3D Medical Image Segmentation Using SAM-Inspired Architecture

Yu Xin[1], Gorkem Can Ates[1], and Wei Shao[1]

University of Florida, Gainesville FL 32611, USA
weishao@ufl.edu

**Abstract.** Existing 3D medical image segmentation methods are often constrained by a fixed set of predefined classes or by reliance on manually defined prompts such as bounding boxes and scribbles, which are often labor-intensive and prone to ambiguity. To address these limitations, we present a framework for 3D medical image segmentation across diverse modalities guided solely by free-text descriptions of target anatomies or diseases. Our solution is built on a multi-component architecture that integrates efficient feature encoding via decomposed 3D convolutions and self-attention, multi-scale text-visual alignment, and a SAM-inspired mask decoder with iterative refinement. The model is further conditioned through a prompt encoder that transforms language and intermediate visual cues into spatially aligned embeddings. To train and evaluate our model, we used a large-scale dataset of over 200,000 3D image-mask pairs spanning CT, MRI, PET, ultrasound, and microscopy. Our method achieved an average Dice of 0.609 and F1 score of 0.113 on the open validation set, outperforming baselines such as CAT (Dice 0.532, F1 0.194) and SAT (Dice 0.557, F1 0.096). It showed strong generalization across modalities, with particularly high performance on ultrasound (Dice 0.829) and CT (Dice 0.672). These results confirm the feasibility of free-text-guided 3D segmentation and establish our approach as a strong foundation model for general-purpose medical image segmentation. Our code is publicly available at: https://github.com/mirthAI/Text3DSAM/.

**Keywords:** Text-guided segmentation · 3D medical imaging · Vision-language models.

## 1 Introduction

Medical image segmentation is essential for clinical tasks such as diagnosis, treatment planning, and quantitative analysis. Traditional models rely on supervised learning with fixed class labels and voxel-wise manual annotations, which are costly to obtain, difficult to scale, and unable to generalize to new tasks or unseen structures without retraining. These challenges are compounded by the high variability in medical images due to differences in scanners, acquisition protocols, and patient anatomy, as well as the limited size and diversity of labeled datasets. Moreover, producing accurate annotations requires expert knowledge, adding further barriers to scalability in real-world clinical settings.

To address the limitations of traditional segmentation methods, prompt-based approaches have emerged as a more flexible alternative. Instead of predicting all classes at once, these models can segment a specific structure based on user input. This focused interaction improves accuracy and enables a single model to adapt to a variety of tasks without retraining. Foundation models like the Segment Anything Model (SAM) [6] and its successor SAM2 [12] enable prompt-based segmentation in natural images using minimal supervision via points or boxes. Inspired by these advances, medical adaptations such as MedSAM [8] and MedSAM2 [10] have extended these architectures to 3D medical imaging data. However, these models rely heavily on spatial prompts, limiting their usability in clinical workflows where generating such inputs is labor-intensive or ambiguous. Moreover, they lack support for more expressive and intuitive forms of interaction, such as natural language. To address these issues, recent efforts have introduced text-guided segmentation frameworks such as BioMedParse [17], CAT [4], and SAT [18], which leverage descriptive textual prompts for semantic control.

We propose **Text3DSAM**, a text-guided segmentation framework for 3D medical images across diverse modalities. Rather than relying on spatial prompts like bounding boxes or points, Text3DSAM accepts natural language descriptions to define target anatomies or pathologies, enabling more intuitive interaction, reduced annotation effort, and zero-shot generalization to unseen classes. The architecture integrates decomposed 3D convolutions and self-attention for efficient image encoding, multi-scale alignment between textual and visual features, and a SAM-inspired mask decoder for high-quality segmentation. Trained on a large-scale dataset of over 200,000 3D image-mask pairs spanning CT, MRI, PET, ultrasound, and microscopy, our model demonstrates strong generalization and outperforms state-of-the-art baselines such as CAT and SAT, particularly on CT and ultrasound.

## 2    Method

We propose **Text3DSAM**, a text-guided segmentation framework for 3D medical imaging across multiple modalities. Given a volumetric scan and a natural language prompt (e.g., "liver in abdominal CT"), our model outputs a high-quality segmentation mask for the specified region. As illustrated in Fig. 1, Text3DSAM comprises four core components: an 3D image encoder for volumetric feature extraction, a text encoder for semantic embedding, a prompt encoder for segmentation-aware conditioning, and a SAM-inspired mask decoder for mask generation and refinement.

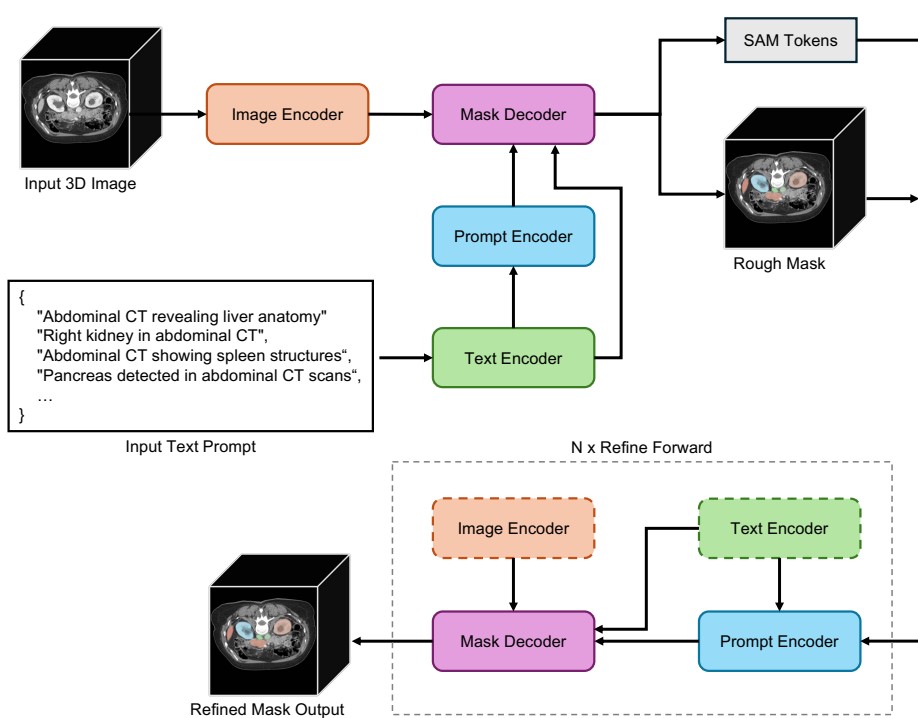

**Fig. 1.** Overview of Text3DSAM.

## 2.1  Efficient Image Encoder

Our image encoder builds on DCFormer [1], an efficient 3D encoder architecture based on decomposed 3D convolutions. DCFormer has demonstrated superior performance over standard 3D vision transformers in tasks such as zero-shot disease detection [1], image-text retrieval [1,15], radiology report generation, and visual question answering [15]. While DCFormer uses large convolutional kernels to capture global context, we further enhance its feature modeling by appending a standard 3D transformer layer after the final decomposed convolution block. The modified image encoder is illustrated in Fig. 2.

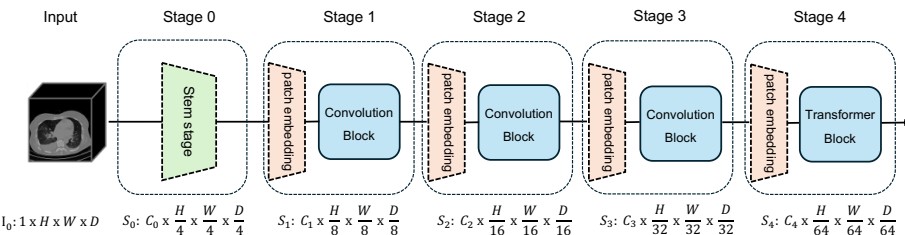

**Fig. 2.** Architecture of the proposed 3D Image Encoder.

## 2.2   Prompt Encoder

The Prompt Encoder (see Fig.3) transforms user-provided text prompts and optional visual hints into sparse and dense embeddings that guide the segmentation process. It comprises a positional encoding module, a 3D convolution-based mask encoder, and a sparse embedding aggregator. This design is inspired by the prompt encoding mechanisms in SAM [6] and SegVol [3], which encode both semantic and spatial cues to enhance downstream mask prediction.

During the initial forward pass, the encoder operates using only the input text embeddings. A learnable no-mask embedding is used to construct the dense embedding, which is broadcast to align with the spatial dimensions of the image features. This setup enables the model to produce a coarse segmentation prediction based solely on the semantic intent of the text, without relying on any spatial guidance. In subsequent refinement stages, the Prompt Encoder incorporates two additional inputs: a rough segmentation mask and SAM tokens generated by the previous Mask Decoder. The rough mask is processed by a lightweight 3D convolutional encoder composed of DecompConv blocks from DCFormer [1], which extract multiscale spatial features and produce a dense embedding aligned with the image feature space. At the same time, the sparse embedding is updated by concatenating the previous SAM tokens with the text embedding, allowing the model to integrate both semantic intent and visual feedback.

This two-stage design supports iterative, coarse-to-fine refinement of the predicted masks. Early stages depend primarily on semantic guidance from the text prompt, while later stages progressively incorporate spatial and contextual cues from intermediate outputs. This approach enhances the model's ability to produce accurate and contextually aligned segmentations across diverse anatomical and imaging scenarios.

To maintain spatial consistency during these stages, all dense embeddings are augmented with positional encodings generated by a randomized Fourier-based module. These encodings are shared across the batch and passed to the Mask Decoder to guide attention computations within the transformer. This mechanism ensures that the embeddings preserve spatial alignment throughout the segmentation pipeline.

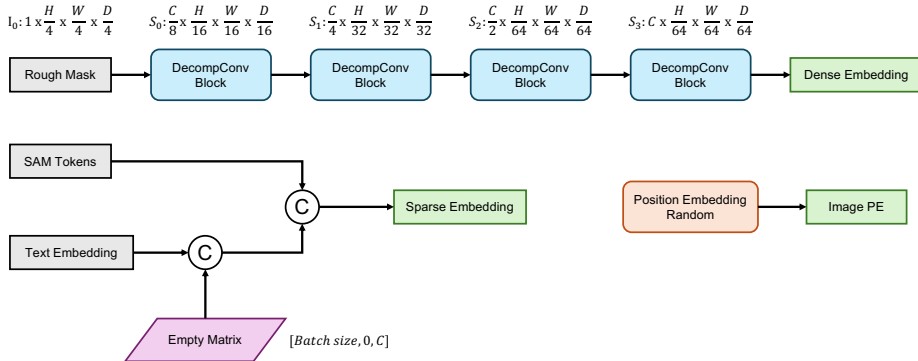

**Fig. 3.** Architecture of the proposed prompt encoder.

## 2.3   Mask Decoder

The Mask Decoder predicts segmentation masks conditioned on both visual features and prompt embeddings. As illustrated in Fig. 4, it consists of a transformer-based core and a two-stage mask generation pipeline that enables semantic-region alignment in volumetric space. The design draws inspiration from SAM [6] for token-based decoding and SegVol [3] for multimodal feature alignment, while incorporating high-resolution skip connections similar to U-Net [14] and SAM2 [12].

At each refinement stage, the decoder receives image embeddings, sparse and dense prompt embeddings, and positional encodings generated by the Prompt Encoder. A set of learnable mask tokens is concatenated with the sparse prompt tokens and passed into a transformer, which jointly processes the token sequence and image features through attention. The transformer outputs updated mask tokens and refined spatial features, which serve as the basis for generating the segmentation masks.

To reconstruct high-resolution masks, the spatial features are upsampled in two stages using trilinear interpolation and lightweight $1\times1\times1$ convolutions. When available, high-resolution features are fused with the upsampled features via residual addition and shallow convolutions. This design—combining U-Net-style skip connections and SAM2-inspired feature fusion—enhances both boundary localization and volumetric continuity, especially for fine anatomical structures.

Each mask token is passed through a lightweight MLP-based hypernetwork that generates dynamic convolution kernels, which are used to project the refined spatial features into binary mask logits. To further enhance semantic alignment, a cross-modal similarity mechanism inspired by SegVol [3] is introduced. Specifically, the text embedding is linearly projected and compared against the upsampled feature volume, and the resulting similarity map is added to each predicted mask to guide the model toward regions matching the semantic prompt.

Only the first predicted mask is retained as the final output at each stage, and its associated token is returned to the Prompt Encoder for the next refinement step. This iterative process allows the model to progressively improve its segmentation predictions in a coarse-to-fine manner, integrating feedback from earlier stages to refine both spatial precision and semantic consistency.

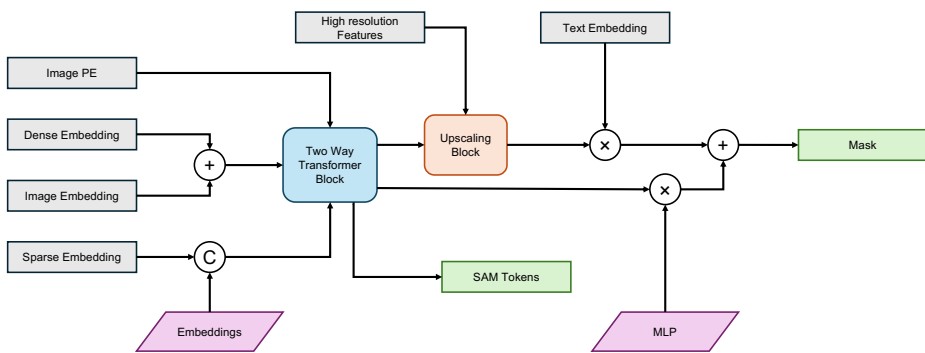

**Fig. 4.** Architecture of the proposed mask decoder.

### 2.4   Loss Function

We adopt a hybrid loss function that combines Dice loss and Focal loss to jointly optimize region-level overlap and voxel-wise accuracy. This combination balances precision in segmentation overlap with robustness to class imbalance, which is especially important in volumetric medical segmentation. Dice loss encourages accurate region-level alignment and is defined as:

$$\mathcal{L}_{\text{Dice}} = 1 - \frac{2\sum_i \hat{y}_i y_i + \epsilon}{\sum_i \hat{y}_i + \sum_i y_i + \epsilon} \tag{1}$$

where $\epsilon$ is a small constant for numerical stability, $\hat{y} \in [0,1]$ is the predicted probability volume, and $y \in 0,1$ is the corresponding ground truth mask.

Focal loss emphasizes hard-to-classify voxels by down-weighting the contribution of easy examples:

$$\mathcal{L}_{\text{Focal}} = -\sum_i \alpha(1 - \hat{y}_i)^\gamma y_i \log(\hat{y}_i) \tag{2}$$

where $\alpha \in [0,1]$ is a balancing factor, and $\gamma > 0$ is a focusing parameter that reduces the relative loss for well-classified voxels.

The final loss is the sum of both terms:

$$\mathcal{L}\text{total} = \mathcal{L}\text{Dice} + \mathcal{L}_{\text{Focal}} \tag{3}$$

This combined objective promotes both global mask accuracy and fine-grained sensitivity to subtle or underrepresented regions.

## 3    Experiments

### 3.1    Dataset and evaluation metrics

The development set is an extension of the CVPR 2024 MedSAM on Laptop Challenge [9], including additional 3D cases from public datasets[1] and covering commonly used 3D modalities such as Computed Tomography (CT), Magnetic Resonance Imaging (MRI), Positron Emission Tomography (PET), Ultrasound, and Microscopy. The hidden testing set was created through a community effort, with all cases being unpublished. Annotations were either provided by data contributors or generated by the challenge organizers using 3D Slicer [5] and MedSAM2 [10]. In addition to the full-data track, the challenge also includes a coreset track, in which participants may use only 10% of the training cases for model development.

The text-guided segmentation task includes both semantic segmentation and instance segmentation. For semantic segmentation, evaluation metrics include the Dice Similarity Coefficient (DSC) and Normalized Surface Distance (NSD), which assess region overlap and boundary accuracy, respectively. For instance segmentation, we compute the F1 score at an overlap threshold of 0.5 and report the DSC for true positives. Additionally, algorithm runtime is limited to 60 seconds per class; exceeding this limit results in all DSC and NSD metrics being set to zero for that test case.

### 3.2    Implementation details

**Preprocessing**  Following the practice in MedSAM [8], all images were converted to npz format and rescaled to an intensity range of $[0, 255]$. For CT images, we first normalized Hounsfield units using standard window width and level settings: soft tissues (W:400, L:40), lung (W:1500, L:-160), brain (W:80, L:40), and bone (W:1800, L:400). The normalized intensities were then linearly rescaled to $[0, 255]$. For other modalities, intensity values were clipped between the 0.5th and 99.5th percentiles before rescaling to the same range. If an image was already within $[0, 255]$, no further preprocessing was applied.

To ensure data integrity, we traversed the entire dataset to verify that each file could be opened successfully and that every sample contained both an image and its corresponding mask. For each valid sample, we recorded the file path and all class IDs present in both the segmentation mask and the associated text prompt. This metadata was stored in a JSON file and later used during dataset construction.

**Environment settings**  The development environments and requirements are presented in Table 1.

---

[1] A complete list is available at https://medsam-datasetlist.github.io/

**Table 1.** Development environments and requirements.

| | |
|---|---|
| System | RHEL 8 |
| CPU | 16×8 cores of AMD EPYC 7742 64-Core Processor |
| RAM | 16×128GB |
| GPU (number and type) | 16× NVIDIA A100 80GB SXM |
| CUDA version | 12.4.1 |
| Programming language | Python 3.12 |
| Deep learning framework | torch 2.7.0, deepspeed 0.16.5, huggingface-hub 0.30.1 |

**Training protocols** All images and masks were resized to a uniform resolution of $128 \times 256 \times 256$ using trilinear and nearest-neighbor interpolation, respectively. To increase spatial diversity, we applied random 90-degree rotations along in-plane dimensions, followed by intensity perturbations using `RandScaleIntensity` and `RandShiftIntensity`. Random flipping was intentionally excluded, as many segmentation prompts reference anatomically lateralized organs (e.g., left/right lung), where flipping could introduce semantic ambiguity.

For each sample, the class ID was used to retrieve corresponding natural language descriptions from a predefined text prompt file, from which one prompt was randomly selected at runtime to serve as the semantic query. We used a pre-trained language model, TinyClinicalBERT [13], to encode the text prompts. The model processes each input volume—resized to $128 \times 256 \times 256$—by generating an initial coarse prediction followed by two refinement iterations (`pass_num` = 2), with intermediate masks and tokens fed back into the network for progressive improvement.

Our training pipeline was implemented using PyTorch [11] and MONAI [2], with DeepSpeed used to support scalable, efficient model training. All experiments were conducted using BF16 mixed-precision to reduce memory usage and improve performance. The model was trained for 30 epochs using the AdamW [7] optimizer with a base learning rate of $1 \times 10^{-4}$, weight decay of $1 \times 10^{-5}$, and a cosine learning rate schedule with a warm-up ratio of 0.03. We used a batch size of 16 per device and employed 8 data loader workers to optimize I/O throughput during training.

**Table 2.** Training protocols.

| | |
|---|---|
| Batch size | 16 |
| Patch size | $128{\times}256{\times}256$ |
| Total epochs | 30 |
| Optimizer | AdamW |
| Initial learning rate (lr) | $1 \times 10^{-4}$ |
| Lr decay schedule | cosine |
| Training time | 15 hours |
| Loss function | Dice Loss + Focal Loss |
| Number of model parameters | 59.30 M[2] |
| Number of flops | 48.34 G (inference) / 145.02 G (training)[3] |

## 4  Results and discussion

We present both qualitative and quantitative results to evaluate our model's performance across multiple modalities and segmentation tasks, highlighting its strengths and current limitations.

### 4.1  Qualitative results on validation set

Figure 5 shows representative cases where the model succeeds. These typically involve large, clearly visible, and semantically distinct regions. Predictions in these regions closely match the prompts and demonstrate accurate shapes and boundaries. Prompts referencing anatomically well-defined structures, such as "liver in CT" or "ventricles in MRI," are more reliably interpreted and lead to better segmentation.

Figure 6 illustrates several common failure modes. In some cases, low image quality or contrast—particularly in Microscopy or PET scans—prevents the model from capturing clear boundaries. In others, very small target regions combined with the limited resolution of the mask decoder result in missed predictions or coarse approximations. Some failures occur when the segmented region does not match the intended anatomical structure, indicating that the model misinterpreted the text prompt. These errors highlight current limitations in multimodal alignment and reasoning under sparse supervision.

### 4.2  Quantitative results on validation set

Table 3 summarizes the performance of different methods on the validation set under the all-data track. Text3DSAM outperforms baseline methods (CAT and SAT) across most modalities in terms of both Dice Similarity Coefficient (DSC) and F1_50 scores.

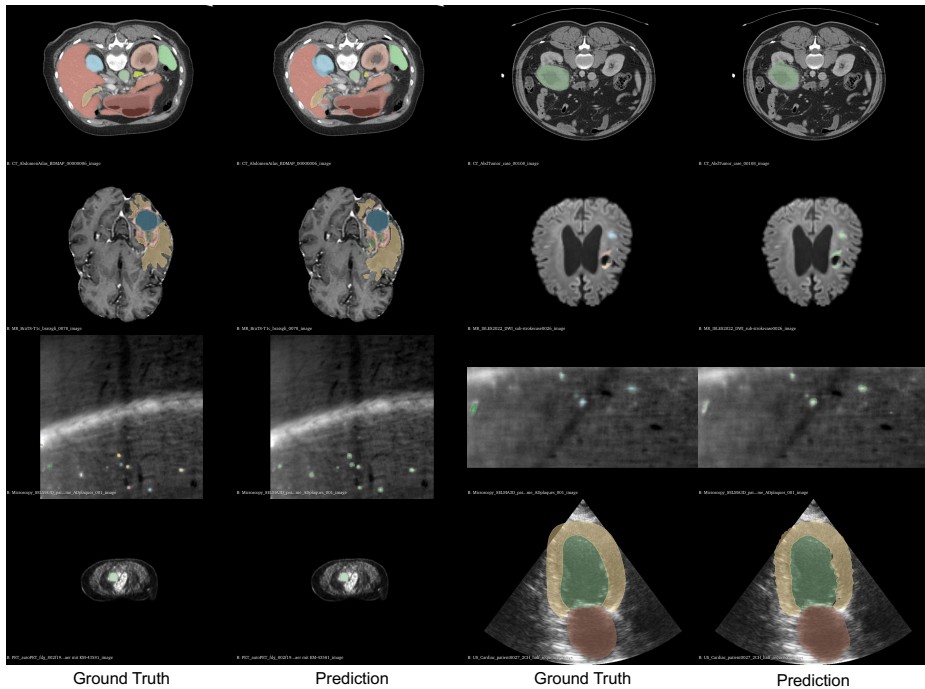

Ground Truth      Prediction      Ground Truth      Prediction

**Fig. 5.** Qualitative examples of successful segmentations from the validation set on the **all-data track**.

In semantic segmentation, Text3DSAM shows clear improvements, achieving the highest DSC scores across all cases. This suggests our model effectively captures global context and produces spatially coherent segmentations when guided by clear textual prompts. For example, it achieves a DSC of 0.6707 on CT and 0.8337 on Ultrasound, demonstrating strong performance on modalities with well-defined structures.

In contrast, the model's performance on instance segmentation remains limited. In Microscopy and PET, which require precise instance-level delineation and boundary accuracy, SAT occasionally outperforms Text3DSAM. This suggests the mask decoder still struggles with fine-grained separation of small or overlapping structures, indicating a need for higher-resolution output and improved instance-level reasoning.

### 4.3   Limitation and future work

While Text3DSAM achieves strong performance on many semantic segmentation tasks, its ability to handle instance-level segmentation and small object detection remains limited. The coarse mask resolution constrains the precision required to

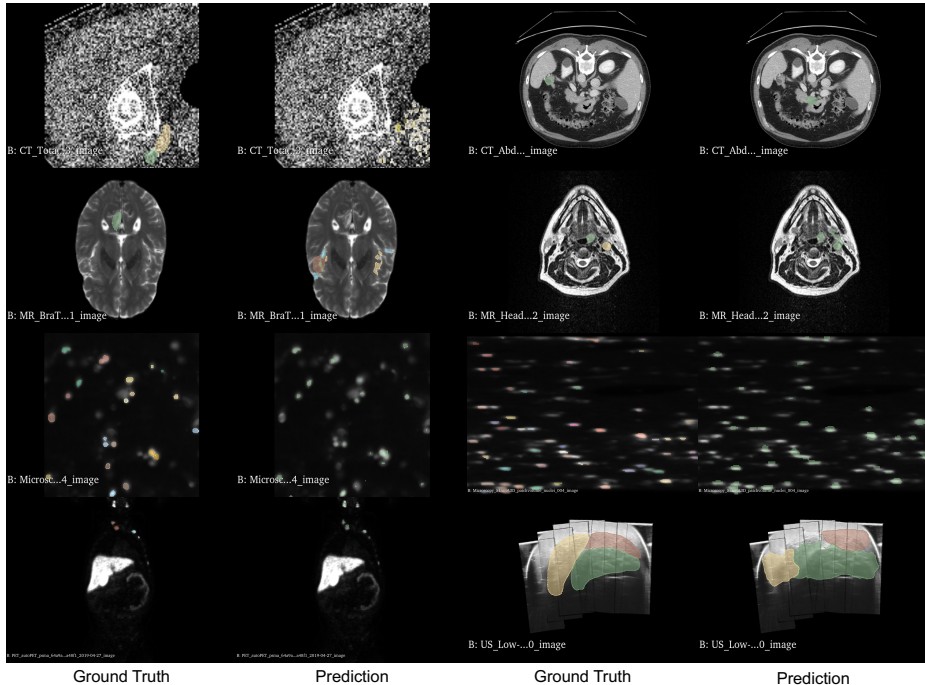

**Fig. 6.** Qualitative examples of failure cases from the validation set on the **all-data track**.

delineate tiny regions, and the current multimodal fusion design does not always ensure accurate interpreation of the prompt.

Future work will focus on enhancing the resolution and expressiveness of the mask decoder to capture finer details and better resolve overlapping structures. Improving semantic fusion between vision and language inputs will also be a priority, particularly for prompts involving complex spatial or instance-level cues. Furthermore, adopting a multiscale decoding strategy may may improve performance on targets of varying sizes, especially small or sparse regions that are currently challenging for the model.

## 5   Conclusion

We propose **Text3DSAM**, a flexible and generalizable framework for text-guided 3D medical image segmentation. It leverages prompt-based supervision to generate anatomically consistent masks without requiring manual annotations. The model demonstrates strong performance across multiple modalities, particularly in semantic segmentation. Despite current limitations in instance segmentation and small object detection, the results indicate promising gener-

**Table 3.** Quantitative evaluation results of the validation set on the **all-data track**.

| Modality | Method | DSC | F1_50 |
|---|---|---|---|
| CT | CAT | 0.6035 | 0.2573 |
|  | SAT | 0.6432 | 0.1032 |
|  | Text3DSAM | 0.6707 | 0.1148 |
| MRI | CAT | 0.4255 | 0.1511 |
|  | SAT | 0.4526 | 0.0373 |
|  | Text3DSAM | 0.5214 | 0.1202 |
| Microscopy | CAT | N/A | 0.0211 |
|  | SAT | N/A | 0.2475 |
|  | Text3DSAM | N/A | 0.0386 |
| PET | CAT | N/A | 0.1106 |
|  | SAT | N/A | 0.2623 |
|  | Text3DSAM | N/A | 0.0886 |
| Ultrasound | CAT | 0.8180 | N/A |
|  | SAT | 0.7549 | N/A |
|  | Text3DSAM | 0.8337 | N/A |
| Average | CAT | 0.5316 | 0.1935 |
|  | SAT | 0.5573 | 0.0956 |
|  | Text3DSAM | 0.6091 | 0.1131 |

alization and multimodal reasoning capabilities. Future work will focus on improving mask resolution and semantic alignment to further enhance performance across diverse segmentation tasks.

**Acknowledgements** We thank all the data owners for making the medical images publicly available and CodaLab [16] for hosting the challenge platform. This work was supported by the Department of Medicine and the Intelligent Clinical Care Center at the University of Florida College of Medicine. The authors express their sincere gratitude to the NVIDIA AI Technology Center at the University of Florida for their invaluable feedback, technical guidance, and support throughout this project.

**Disclosure of Interests.** The authors have no competing interests to declare that are relevant to the content of this article.

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

**Table 4.** Checklist Table. Please fill out this checklist table in the answer column. (**Delete this Table in the camera-ready submission**)

| Requirements | Answer |
|---|---|
| A meaningful title | Yes |
| The number of authors (≤6) | 3 |
| Author affiliations and ORCID | Yes |
| Corresponding author email is presented | Yes |
| Validation scores are presented in the abstract | Yes |
| Introduction includes at least three parts: background, related work, and motivation | Yes |
| A pipeline/network figure is provided | Figure 1, 2, 3, 4 |
| Pre-processing | Page 7 |
| Strategies to data augmentation | Page 8 |
| Strategies to improve model inference | Page 8 |
| Post-processing | None |
| Environment setting table is provided | Table 1 |
| Training protocol table is provided | Table 2 |
| Ablation study | Page 9 |
| Efficiency evaluation results are provided | Table 3 |
| Visualized segmentation example is provided | Figure 5, 6 |
| Limitation and future work are presented | Yes |
| Reference format is consistent. | Yes |
| Main text >= 8 pages (not include references and appendix) | Yes |