# OpenReview forum: "Text3DSAM: Text-Guided 3D Medical Image Segmentation Using SAM-Inspired Architecture"
_thecvf.com/CVPR/2025/Workshop/MedSegFM — CVPR 2025 Workshop MedSegFM Submission_

### Official Review · Reviewer_9Xxu · 2025-09-27
**A Promising Text-Guided Foundation Model for 3D Medical Segmentation**

**Rating:** 7
**Confidence:** 5

**Review:**

This paper introduces Text3DSAM, an innovative and robust framework for text-guided 3D medical image segmentation across diverse modalities. By using natural language descriptions instead of traditional spatial prompts, the model significantly improves usability for general-purpose segmentation. The architecture, which integrates an efficient 3D encoder, multi-scale text-visual alignment, and an iterative, SAM-inspired mask decoder (with SAM2-inspired fusion), is technically sound. Text3DSAM achieves substantial quantitative improvements, particularly in Dice score, over state-of-the-art baselines (CAT and SAT). The strong results, clear methodology, and public code release make this a valuable contribution to the field. While limitations remain in instance segmentation and certain design choices could benefit from further ablation, the work warrants acceptance.

Strengths:
1. The model achieves a notable boost in overall performance, with an average Dice score of 0.609, significantly outperforming baselines (SAT: 0.557, CAT: 0.532). The gains are particularly strong on well-defined structures in modalities like CT and Ultrasound.
2. The SAM2-inspired feature fusion within the Mask Decoder is a key innovation, effectively combining U-Net skip connections to enhance boundary localization and volumetric continuity for fine anatomical structures.
3. The core idea of purely text-guided 3D segmentation across diverse modalities is a crucial step towards flexible foundation models in medicine.
4. The paper features clear explanations, explicit quantitative results, and a commitment to public code release.

Weaknesses:
1. The model's performance on instance segmentation remains limited, particularly for small or overlapping structures in modalities like Microscopy and PET. This is likely constrained by the current mask decoder and the fixed input size of 128×256×256. Future work must focus on higher-resolution output and improved instance-level reasoning.
2. The choice of two refinement iterations (pass_num=2) needs justification. A simple ablation study comparing pass_num=1,2,and 3 is recommended to validate this design choice.
3. The feature fusion involving high-resolution features in the UpscalingBlock is a vital design element. A concise comparison quantifying the performance impact of including vs. excluding these high-resolution features would be valuable to demonstrate their contribution to boundary accuracy. The Fig. 4. should be updated to more clearly show where these high-resolution features are fused.

---

> ### Author Response · Authors · 2025-11-06
>
> We sincerely thank the reviewer for the constructive comments and the positive assessment of our work. We have carefully considered the points raised and provide our responses below:
>
> 1. We acknowledge that reshaping all modalities to a fixed size of 128×256×256 may negatively impact certain modalities, particularly for small or overlapping structures. We plan to address this limitation in future work by exploring higher-resolution inputs and improved instance-level reasoning.
> 2. We agree that additional ablation studies would provide deeper insight into the contributions of different components, including the number of refinement iterations (pass\_num) and the effect of high-resolution feature fusion. We plan to include these analyses in future work.
> 3. We will update the figures in the manuscript, particularly Figure 4, to more clearly illustrate where high-resolution features are fused and help readers better understand the model architecture.
>
> We thank the reviewer again for the valuable feedback, which will guide future improvements and enhance the clarity of our work.

---

### Official Review · Reviewer_mRAu · 2025-09-29
**Text3DSAM: Text-Guided 3D Medical Image Segmentation Using SAM-Inspired Architecture**

**Rating:** 6
**Confidence:** 4

**Review:**

This paper presents Text3DSAM, a text-driven 3D medical image segmentation framework built upon the Segment Anything architecture. It addresses the limitations of manually defined prompts (such as bounding boxes and scribbles, which are often labor-intensive and prone to ambiguity) by enabling segmentation through free-text descriptions of target anatomies or diseases. Experimental results demonstrate that Text3DSAM significantly outperforms baseline methods, validating its effectiveness. The authors further strengthen their contribution by releasing detailed documentation and publicly available code, thereby enhancing reproducibility and supporting future research in the field.

Strengths:
- The core contribution of this paper is enabling the Segment Anything Model (SAM) to support text prompts in 3D medical image segmentation, effectively removing reliance on spatial prompts like bounding boxes or scribbles.
- The authors introduce a modified version of DCFormer by appending a standard 3D transformer block after the final decomposed convolution layer. This architectural change enhances the model’s ability to capture complex spatial-text relationships.
- The proposed method demonstrates improved performance over baseline models on the validation dataset, reinforcing the effectiveness of the approach.

Weaknesses:
- Despite multiple architectural modifications, the paper does not include ablation studies to isolate the impact of each component. This omission makes it difficult to determine which changes contribute most to the performance gains.
- In Figure 2, the addition of a transformer block to the end of DCFormer raises questions. The authors do not explain why this block is appended rather than replacing all convolutional blocks or why it achieves higher performance than vanilla DCFormer.
- The paper lacks a clear definition of its main contributions. Readers may struggle to distinguish the novelty of this work from prior methods, which weakens its overall impact.
- Figures contain numerous undefined terms (e.g., MLP, PE) that should be clarified at least once. Figure 4, in particular, includes labels like “Sparse Embedding,” “High Resolution Features,” and “Text Embedding” without indicating their origin, whether from the encoder, user prompt, or another module. This lack of clarity hinders understanding.
- The role of SAM tokens in the mask decoder is unclear. If these tokens are not essential, they should be removed (e.g., from my knowledge, these tokens should be multiplied by MLP outputs to generate masks). In addition, the authors should explain the input to the MLP (e.g., a learnable mask token).
- The injection of text embeddings after the upscaling block is quite confusing, where embeddings are typically introduced via prompt tokens and two-way transformers (from SAM). The authors should provide a clear motivation and explanation for this design choice, especially regarding how masks are generated and why text embeddings are needed at the final stage.

Overall, the primary limitation of this paper lies in its lack of clarity regarding the specific problems each proposed module is designed to address. While the model demonstrates improved performance over baseline methods, the paper does not sufficiently explain why these modules contribute to that improvement. Without clear motivation, functional justification, or supporting ablation studies, it becomes difficult for readers to understand the role and impact of each component within the framework.

---

> ### Author Response · Authors · 2025-11-06
>
> We sincerely thank the reviewer for the detailed and constructive feedback. We have carefully considered each point and provide our responses below:
>
> 1. We acknowledge the absence of ablation studies isolating the contribution of each component. Due to resource constraints, we were unable to include them in the current version. We plan to explore comparisons across different image encoders, feature fusion settings, and iterative refinement passes in future work to better understand each module’s contribution.
> 2. The appended Transformer block at the final layer of DCFormer is motivated by the relatively low resolution of 3D input volumes. After multiple decomposed convolution blocks, the feature map becomes small, allowing the Transformer block to enhance global feature representation without increasing parameters or computation significantly.
> 3. We will revise the manuscript to provide a clear, concise definition of the main contributions to distinguish novelty from prior work.
> 4. We will update the manuscript to clarify all figure labels (e.g., MLP, PE, High Resolution Features, Sparse Embedding, Text Embedding) and their origins, and ensure that diagrams clearly indicate data flow and module interactions.
> 5. In our multi-pass pipeline, SAM tokens from the previous pass serve as inputs to the next pass’s prompt encoder. This design helps the two-way transformer better capture segmentation region information across passes.
> 6. Text embeddings are injected after the Upscaling Block to maintain semantic alignment with the prompt. Features tend to lose prompt-related information after multiple upscaling operations. Unlike contrastive loss approaches, which are computationally expensive, our lightweight matrix multiplication of text embeddings with upscaled features preserves prompt adherence with minimal overhead, improving the quality of the final segmentation mask.
>
> We thank the reviewer again for the valuable comments, which will help improve clarity, reproducibility, and the overall quality of our work.

---

### Official Review · Reviewer_dVuv · 2025-10-10
**Good contribution to text guided segmentation for 3d medical imaging.**

**Rating:** 6
**Confidence:** 4

**Review:**

Promising and clinically relevant step toward text guided 3D segmentation with good performance over baselines and an efficient design. However, no backbone ablations for image encoder, potentially uncalibrated decoder fusion, and text encoder is not specified whether frozen or fine-tuned and how it is projected with prompt encoder.

# Strengths
- Targets text guided 3D segmentation that adds high-value for real clinical workflows.
- Uses an efficient 3D image encoder DCFormer to keep 3D compute and memory practical.
- Outperforms baselines methods on validation set, indicating the approach is effective in practice.

# Weaknesses
- Lacks ablations vs. standard 3D CNN/Transformer backbones, so the choice of DCFormer isn’t fully known for this task.
- Plain fusion in the mask decoder can cause scale mismatch, might need normalization or other calibration.
- With TinyClinicalBERT, it’s unclear if it’s frozen vs. fine-tuned, how the text vector is projected, and where and how it is injected.

---

> ### Author Response · Authors · 2025-11-06
>
> We sincerely thank the reviewer for the positive assessment of our work and the constructive feedback. We have carefully addressed each concern as follows:
>
> 1. We acknowledge the absence of ablation studies comparing DCFormer with standard 3D CNN and Transformer backbones. Due to time and resource constraints, these experiments were not included in the current version. We plan to explore this comparison, along with analyses of feature fusion and iterative refinement, in our future work to better understand each component’s contribution.
> 2. After obtaining the text embeddings, we employ an MLP block to project them into the same hidden dimension as the image features. In the mask decoder, the original text embedding further passes through another MLP to match the upscaled image feature dimension before fusion. This design effectively aligns multi-modal features and maintains efficiency. We will clarify this mechanism in the revised manuscript.
> 3. TinyClinicalBERT is jointly fine-tuned with the rest of the model during training to adapt to the clinical domain. We will explicitly state this in the updated version for completeness.
>
> We thank the reviewer again for the insightful feedback, which will guide improvements in both clarity and completeness of the manuscript.

---

### Comment · Reviewer_ZMTk · 2025-10-13
**Meta-Review: Major Revision required for Architectural Clarification and missing Ablations**

The paper presents a clinically relevant text-guided 3D segmentation model with strong quantitative performance over baselines. However, consensus among reviewers is that the manuscript suffers severely from a lack of ablation studies and unclear methodological descriptions, making it difficult to assess the true contribution of specific design choices. A Major Revision is strongly recommended.
The authors must address the following key issues:
1. Severe Lack of Ablation Studies
Despite introducing multiple architectural modifications, the paper fails to isolate the impact of these individual components.
Action Required: Provide ablation studies to quantify the contribution of:
Image Encoder: Compare the proposed "Modified DCFormer" against the original DCFormer and standard 3D backbones (e.g., 3D U-Net or standard ViT).
Feature Fusion: Quantify the performance impact of including vs. excluding high-resolution feature fusion in the mask decoder.
Iterative Refinement: Justify the choice of pass_num=2 by comparing performance across pass_num=1, 2, and 3.
2. Architectural Clarity and Motivations
Several technical details are ambiguous, and the rationale behind specific design choices is missing.
Action Required:
DCFormer Modification: Explain the motivation for appending a Transformer block to the end of DCFormer rather than other integration strategies.
Text Embedding Injection: Clarify why text embeddings are added after the Upscaling Block. This deviates from standard SAM-style fusion (usually via two-way transformers); the motivation for this specific design to ensure semantic alignment must be justified.
Decoder Mechanism: Explicitly define the role and origin of "SAM tokens" in the decoder. What are the inputs to the MLP? How do they generate the mask?
Text Encoder: State clearly if TinyClinicalBERT is frozen or fine-tuned.
3. Presentation and Contributions
Action Required:
Add a clear, bulleted list of specific contributions in the introduction to distinguish novelty from prior work.
Overhaul diagrams (especially Figure 4). Define all abbreviations (e.g., MLP, PE) in captions. Clearly indicate the origin of all inputs (e.g., where do "High Resolution Features" come from?) and explicitly show where fusion occurs.

---

> ### Author Response · Authors · 2025-11-06
>
> We sincerely appreciate the reviewer’s constructive comments and helpful suggestions. We have carefully addressed the main concerns as follows:
>
> 1. We acknowledge that our current work lacks some component-wise ablation experiments. In future work, we plan to include (1) comparisons across different image encoders (e.g., Modified DCFormer vs. other 3D backbones), (2) evaluations with and without mask feature fusion, and (3) performance comparisons across different iterative refinement passes (pass\_num = 1, 2, 3). We believe these analyses will further strengthen the understanding of each design choice.
> 2. We appended a Transformer block to the final layer because the input 3D volumes have relatively low spatial resolution. After multiple decomposed convolution blocks, the feature map becomes compact, allowing global context modeling via a Transformer block without increasing the computational cost.
> 3. We inject text embeddings after the Upscaling Block to enhance instruction-following capability. As features are upsampled, they tend to lose semantic alignment with the text prompt. Rather than using computationally expensive contrastive loss, we combine upscaled image features with text embeddings through a lightweight matrix multiplication, which improves prompt adherence with minimal overhead.
> 4. The MLP block in the mask decoder functions as a 1×1×1 convolution. The upscaled feature is multiplied by the MLP weights to produce a single-channel 3D segmentation mask.
> 5. We will revise the manuscript to improve figure clarity and captions, ensuring that all abbreviations and data flow (e.g., feature sources and fusion points) are clearly defined.
>
> We thank the reviewer again for the valuable feedback, which will guide the continued refinement of our work.

---

### Decision · Program_Chairs · 2025-11-12

Accept